# From Programs to Poses: Factored Real-World Scene Generation via Learned Program Libraries

**Joy Hsu**
Department of Computer Science
Stanford University
joycj@stanford.edu

**Emily Jin**
Department of Computer Science
Stanford University
emilyjin@stanford.edu

**Jiajun Wu**
Department of Computer Science
Stanford University
jiajunwu@cs.stanford.edu

**Niloy J. Mitra**
Department of Computer Science
University College London
n.mitra@cs.ucl.ac.uk

## Abstract

Real-world scenes, such as those in ScanNet, are difficult to capture, with highly limited data available. Generating realistic scenes with varied object poses remains an open and challenging task. In this work, we propose FactoredScenes, a framework that synthesizes realistic 3D scenes by leveraging the underlying structure of rooms while learning the variation of object poses from lived-in scenes. We introduce a factored representation that decomposes scenes into hierarchically organized concepts of room programs and object poses. To encode structure, FactoredScenes learns a library of functions capturing reusable layout patterns from which scenes are drawn, then uses large language models to generate high-level programs, regularized by the learned library. To represent scene variations, FactoredScenes learns a program-conditioned model to hierarchically predict object poses, and retrieves and places 3D objects in a scene. We show that FactoredScenes generates realistic, real-world rooms that are difficult to distinguish from real ScanNet scenes.

## 1 Introduction

Real-world scenes are inherently noisy, varied, and lived-in. For instance, chairs are often arranged based on how people interact within a room, and monitors may be angled to face specific seating arrangements. Capturing these nuanced rooms with noisy object poses remains challenging and labor-intensive; hence, high-quality 3D scene datasets such as ScanNet [1] are still scarce. *How can we learn to generate such realistic scenes from limited data?*

Our key insight is that, despite inherent noisiness, indoor scenes retain significant underlying structure based on how rooms were intentionally designed, following social norms and preferences. Chairs are grouped around tables, and coffee tables are positioned by couches. We propose to leverage this (hidden) structure by first generating programmatic layouts that align with the foundational design of rooms, then modeling the realistic variation of lived-in scenes through a pose prediction model for orienting ob-

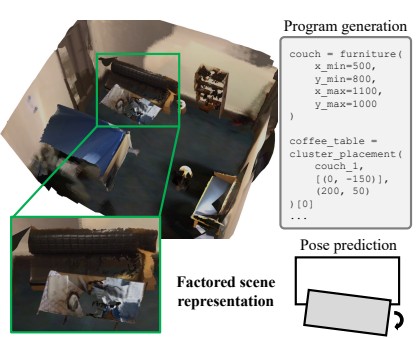

Figure 1: We propose FactoredScenes, a framework that generates layout programs which encode underlying room structure, and predicts objects poses that capture nuanced variation in real-world, lived-in scenes.

39th Conference on Neural Information Processing Systems (NeurIPS 2025).

jects in the room (See Figure 1). Our goal is to synthesize ScanNet-like data with diverse room layouts and object poses.

To this end, we introduce FactoredScenes, a framework that uses a *factored* representation to represent a scene (See Figure 2). We decompose complex scene generation into five steps: (i) learn a library of programs that capture room structures, (ii) generate a scene program using large language models and the learned library, (iii) execute the program to retrieve axis-aligned layouts, (iv) predict object poses with a program-conditioned model, and (v) retrieve object instances based on program structure and predicted dimensions. Notably, such decomposition of a room into hierarchical concepts eliminates the need to directly generate a room sampled from the full scene distribution, learned solely from ScanNet data. Instead, this approach enables FactoredScenes to leverage different data sources and methods to model different components of scenes' structure—effectively bootstrapping learning of the full scene distribution despite limited real-world data. Importantly, this modular design is made possible due to the appropriate levels of abstraction that enable the interface between each component. Semantic knowledge from LLMs is distilled into programs, regularized by learned libraries, which operate on text-parameterized objects, with numeric values predicted by neural networks.

Our framework first learns a space of programs that can generate room layouts from 3D-Front [2], a large-scale dataset of synthetic indoor scenes that are professionally designed. We use this dataset to *learn a library of reusable programs* that captures room structure patterns. FactoredScenes then leverages the generalization capabilities of large language models to create diverse new layouts, guided by our learned library. We demonstrate that this library learning step is essential in capturing structural relationships between objects in rooms, as opposed to relying solely on an inference-based library that has never seen example scenes.

Given the program and generated layout, FactoredScenes learns to predict object poses, using a much smaller ScanNet [1] dataset. Here, the layout program serves as a form of regularization, enabling effective learning from (very) limited real-world data. FactoredScenes's object pose model orients bounding boxes hierarchically given this program. It first predicts poses of primary objects (e.g., a table), and then predicts those of dependent objects (e.g., chairs grouped around the table) based on primary pose predictions. Finally, FactoredScenes retrieves object instances based on their predicted dimensions, completing the full 3D scene.

FactoredScenes demonstrates significant improvements over prior work in generating realistic ScanNet oriented layouts by FID and KID metrics. We also quantitatively evaluate our learned library's ability to compress room structure compared to an inference-only library, and see a $644.1\%$ relative improvement in function use. In addition, we evaluate our object pose model's performance, which shows a $11.4\%$ relative improvement in the prediction of dependent object poses. Finally, we conduct a human study comparing FactoredScenes's rooms to real ScanNet rooms, and demonstrate that our generated 3D scenes are difficult for humans to distinguish from real scenes. We believe that FactoredScenes is a step toward realistic, real-world scene generation from programs to poses.

In summary, our key contributions are the following:

- We propose a library-learning approach to capture the underlying programmatic structure of rooms from 3D-Front.

- We introduce a program-conditioned model for object pose prediction, leveraging hierarchical dependencies between objects and training on limited ScanNet data.

- We validate that FactoredScenes significantly improves upon prior work in generating realistic, oriented layouts.

- We show through human studies that our generated scenes are difficult to distinguish from real ScanNet scenes.

## 2    Related Works

**Indoor scene synthesis.**    A plethora of prior works have been proposed for 3D scene generation [3], with approaches ranging from employing 2D generation models then projecting images to 3D [4, 5, 6, 7], leveraging VAE and GAN architectures with priors [8, 9, 10, 11, 12], and using diffusion-based backbones in a hierarchical and compositional manner [13, 14, 15]. These works also ingest a wide range of input: from using layout as input in 2D image form and scene graph

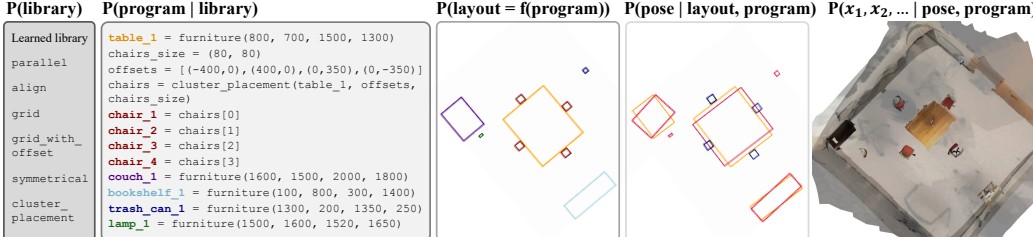

Figure 2: The FactoredScenes framework. FactoredScenes (i) learns a library of programs, (ii) generates a scene program with the learned library, (iii) executes the program to retrieve layouts, (iv) predicts object poses given the program, and (v) retrieves object instances for the full scene. This factorization enables our framework to use different sources of data to generate real-world scenes.

form [10, 16, 17, 18, 19, 20], to using text prompts as input processed by large language models [5, 7, 21, 22, 23].

In this work, we primarily focus on the unconditional scene synthesis task. We highlight four state-of-the-art methods spanning various architectures and designs. ATISS [24] is an autoregressive transformer that predicts plausible room layouts as unordered sets of objects. DiffuScene [25] is a diffusion network that synthesizes 3D indoor scenes by denoising a set of unordered object attributes. Sync2Gen [26] is a variational auto-encoder that learns a latent space of object arrangements. LayoutGPT [27] is a large language model-based planner that generates realistic layouts. Similar to these works, our method first predicts a scene layout and then replaces objects with meshes or point clouds from a set of assets. However, in contrast to these prior works, we focus on the task of real-world unconditional scene synthesis, with *lived-in* scenes. Instead of parameterizing the scene as a collection of axis-aligned labeled bounding boxes, we model each object pose—with not only its bounding box coordinates and size but also its orientation.

**Program-based generation.** Our framework leverages programs as one part of its factored scene representation. Prior works such as Holodeck [21] have also proposed modeling the scene as a constraint-based program, with similar LLM generation of the scene, but across specific modules designed for floorplan, doorway, objects, and layout. Aguina-Kang et al. [22] take a similar high-level approach, with a domain-specific language ingested by the LLM to produce layouts. Other works have also proposed leveraging large language models for scene generation, with templates designed to retrieve structured representations, language to facilitate multi-round interactions [28, 29], and programs and constraints used in various 2D and 3D modeling tasks [30, 31, 27, 32, 33, 34, 35].

Notably, these methods commonly operate on pre-defined languages, which require domain-specific knowledge a priori and cannot flexibly generalize to potential new scene types. They do not see examples of real scenes to *learn* this library from an input dataset. In contrast, FactoredScenes conducts library learning on a large-scale dataset to understand underlying room structure [36]. We show in experiments that our learned program library significantly improves upon an inference-only library in representing scenes compactly. While works such as ShapeCoder [37] follows a similar paradigm, it focuses on training recognition networks for parsing input shapes; in contrast, FactoredScenes uses text-based representations and LLMs to propose and parse named abstractions for downstream generation. The library learning of functions via LLMs enables LLMs to determine the suitable level of abstraction for more precise generation.

## 3 Method

FactoredScenes models rooms as the output of a factorized generative model. We denote a scene $S$ as containing objects $[x_1, x_2, \dots]$, where each object $x$ is represented as an oriented point cloud or mesh. To generate a new scene, we want to sample $S \sim P(S)$. However, learning $P(x_1, x_2, \dots)$ is intractable due to limited real-world data (e.g., ScanNet). Instead, we approximate $P(S)$ with a structured factorization into conditional probabilities (See Figure 2), enabling efficient learning of scene structures and the use of a variety of data sources (e.g., LLMs, synthetic layouts, real scans).

Concretely, we decompose the scene generation process with objects $[x_1, x_2, \dots]$ into the following:

$$P(x_1, x_2, \dots) = P(x_1, x_2, \cdots \mid \text{pose}, \text{program}) \cdot P(\text{pose} \mid \text{layout}, \text{program}) \cdot$$
$$P(\text{layout} = f(\text{program})) \cdot P(\text{program} \mid \text{library}) \cdot P(\text{library}).$$

With this decomposition, we generate real-world scenes in five steps, from programs to poses:

(i) learning a library of programs that capture room structures to model $P(\text{library})$;

(ii) generating a scene program with LLMs regularized by the learned library to model $P(\text{program} \mid \text{library})$;

(iii) executing the program to retrieve layouts to model $P(\text{layout} = f(\text{program}))$;

(iv) predicting object poses with a program-conditioned model to model $P(\text{pose} \mid \text{layout}, \text{program})$;

(v) retrieving object instances based on predicted poses and program structures to model $P(x_1, x_2, \cdots \mid \text{pose}, \text{program})$.

This factorization enables FactoredScenes to learn from different data sources (e.g., library learning of programs from large-scale synthetic data and object pose model training on real-world ScanNet), as well as use appropriate methods for modeling different components (e.g., LLM generation to generalize to new programs and neural networks to predict precise numeric values). FactoredScenes uses LLM-generated programs to determine object locations via commonsense knowledge, and leaves more complex numeric calculations for the pose model trained on real-world orientation variations. Each module leverages its strengths and data available. We describe how we learn each component in the sections below.

## 3.1 Library Learning on 3D-Front

FactoredScenes does not require hand-designed domain-specific functions for generation, but instead *learns* functions through library learning. Its learned library contains functions that capture the structure underlying rooms and specify the possible high-level relationships between objects in a scene; for example, the `cluster_placement` function for grouping objects like chairs around a table. The functions represent reusable layout patterns from which rooms are drawn, which can be used to generate new scenes. We model the library as a discrete uniform distribution over available functions in $L$, where $P(\text{library}) = \text{Uniform}(L)$.

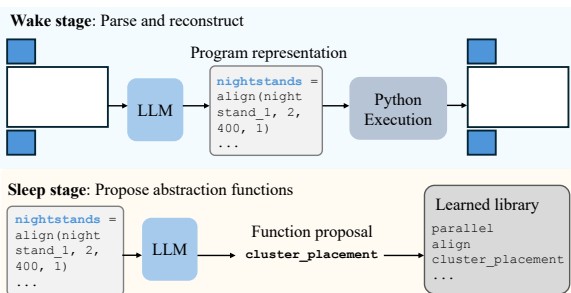

Figure 3: FactoredScenes employs an alternating wake-sleep formulation; in the wake stage, an LLM generates the underlying program of a layout, and in the sleep stage, an LLM proposes new abstractions for the library given successful reconstructions.

Importantly, this library can be learned from synthetic data of professionally designed scenes, without requiring real-world examples, or notably, the need to process complex scenes. Hence, we learn our program library on 3D-Front [2], a large-scale dataset of synthetic indoor rooms with axis-aligned objects. The dataset consists of axis-aligned bounding boxes in a scene (e.g., a layout) and their semantic object labels.

We employ a wake-sleep framework for library learning based on the DreamCoder formulation [36], alternating between generating possible underlying programs and proposing new abstraction functions for the library. Importantly, we propose using a large language model (LLM) as a text-based reasoning model, leveraging the natural parameterization of object bounding boxes into semantic text forms (See Figure 3). In the wake stage, we use an LLM as a recognition model to predict the underlying program given a set of bounding boxes and a library of functions. We bootstrap the library with just two functions: object instantiations and the `parallel` function. The LLM generates the function implementations in Python, and we directly execute the programs as the generation model to output the layout. We then verify whether the predicted programs correctly reconstruct the input bounding boxes, and use it to correct the LLM recognition model and retrieve successful programs.

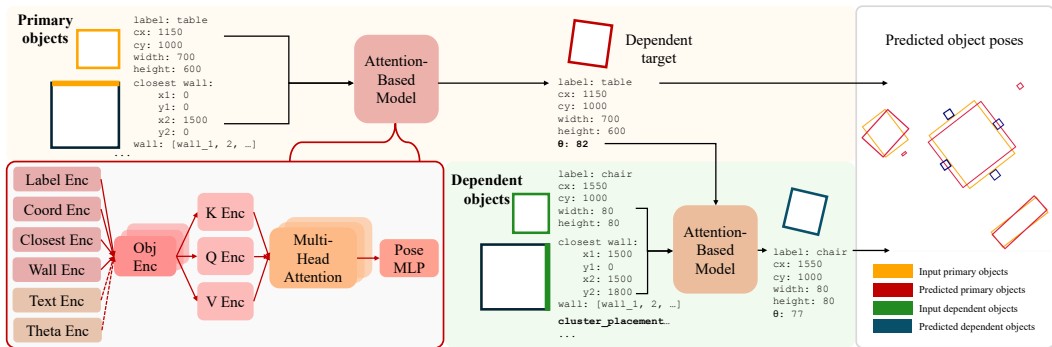

Figure 4: FactoredScenes hierarchically predicts poses for objects based on the underlying program. Our model first predicts primary objects' poses with an attention-based model (e.g., table), then predicts dependent objects' poses additionally conditioned on the orientation and program of their dependency target (e.g., the chairs' poses are dependent on that of table's).

In the sleep stage, given successful programs, we use an LLM as the abstraction proposal model to propose new functions, including their signatures and implementations. We conduct this wake-sleep paradigm iteratively and refine our library for program generation. FactoredScenes discovers the following functions: `align` (for aligning objects like bookshelves in a row), `grid` (for creating a grid of objects like chairs), `grid_with_offset` (for creating a messy grid of objects), `symmetrical` (for placing objects symmetrically around a central point), and `cluster_placement` (for grouping objects like stools around a table). We describe full function signatures for our library in the Appendix.

In this library learning process, we retrieve an abstraction library that captures the underlying structure in rooms, and validate our LLM-based method for parsing the input layout of bounding boxes into programs. We show in ablations that this learned library significantly outperforms an inference-only LLM library, which has never seen example scenes. Notably, compared to prior works that also use LLMs for generation, FactoredScenes's use of LLMs for library learning enables the LLM itself to propose functions at an appropriate level of abstraction for precise generation.

## 3.2 Program Generation via an LLM

Next, we sample $P(\text{program} \mid \text{library})$ by way of an LLM that generates scene programs, conditioned on the learned library and parsed program examples from ScanNet. At a high-level, we distill semantic knowledge in LLMs into program samples, regularized by our library. Concretely, we first parse complex layouts from ScanNet into programs via our library learning framework. We find that the LLM (here, GPT-o1 [38]) can parse layouts into compressed programs that accurately reconstruct the input ScanNet bounding boxes. With these few-shot example programs sampled from ScanNet, the LLM generates new, diverse scene programs.

FactoredScenes then executes the LLM-generated program to retrieve predicted layouts. This execution gives $P(\text{layout} = f(\text{program}))$, where a deterministic Python code interpreter generates layouts consisting of axis-oriented bounding boxes parameterized by their coordinates. Importantly, using an LLM for program generation enables FactoredScenes to generalize to novel scene constructions, while regularizing programs with the space of learned functions and parsed examples enables generation of realistic scenes that follow commonsense rules and structures.

## 3.3 Object Pose Prediction on ScanNet

Given the predicted program and layout of axis-aligned bounding boxes, FactoredScenes learns a model for $P(\text{pose} \mid \text{layout}, \text{program})$. Here, the pose is defined by the orientation $\theta$ for each object in the scene. By predicting oriented bounding boxes, instead of axis-aligned as in most prior work, FactoredScenes is able to capture realistic variation in real, lived-in scenes.

We train our object pose model on ScanNet, a dataset with 707 unique scenes, each with annotated segmented objects. From these rooms, we extract axis-aligned bounding boxes based on the objects' boundaries. To retrieve oriented bounding boxes as ground truth labels for the orientation $\theta$, we

enumerate through 180 degrees of rotation and find the tightest bounding box. Through this process, we generate a small set of real-world examples consisting of axis-aligned bounding boxes and their transformed object pose. For each ScanNet scene, we also retrieve the underlying program via the wake stage of our library learning framework as described.

As there is limited data in ScanNet, we propose leveraging the program structure directly in the forward pass of FactoredScenes's object pose model (See Figure 4). At a high level, FactoredScenes hierarchically predicts object poses for specific objects based on the program context. FactoredScenes first separates objects as *primary* or *dependent* based on the program. Primary objects are those that are directly initialized by global coordinates, while dependent objects are defined by relative relations to a *dependency target*, a previously existing object, through functions in the learned program library. As an example, a table may be a primary object and dependency target that several dependent chair objects may be clustered around. To capture the hierarchical dependency that people move chairs to orient around tables, FactoredScenes first predicts the table's pose, then uses its orientation as input to predict the chairs' poses. All dependent objects' pose predictions are conditioned upon their dependency target as specified by the program.

More concretely, our model predicts the orientation $\theta$ of primary objects, by first encoding the input object label (e.g., *table*), its axis-aligned bounding box (as specified by $x_{min}, y_{min}, x_{max}, y_{max}$), the closest wall (as specified by $x_1, y_1, x_2, y_2$ representing two endpoints as well as its orientation), and all walls in the scene. These embeddings are fused and processed jointly with multi-head self-attention over object slots, then used to predict $\theta$. For dependent objects, in addition to the prior elements, the model also encodes the predicted orientation of the object's dependency target, as well as text embeddings of the program function that instantiated the object (e.g., `chair_1 = cluster_placement(table_1, offsets, (90, 120))`), then passes the result through the same attention stack to predict object poses.

We train FactoredScenes to predict the orientation $\theta$, by treating angles modulo 180° and discretized into 36 classes (5° bins). The loss is cross-entropy over objects, summed over the independent and dependent stages:

$$\mathcal{L} = \mathcal{L}_\theta^{\text{indep}} + \mathcal{L}_\theta^{\text{dep}}; \quad \mathcal{L}_\theta^{\text{stage}} = -\sum_{i,j} \sum_{k=1}^{K} y_{ij,k}^* \log \hat{y}_{ij,k},$$

where $K$ is the number of orientation classes, $\hat{y}_{ij,k}$ is the predicted probability for object $j$ in scene $i$ belonging to class $k$, and $y_{ij,k}^*$ is the one-hot ground truth label. During training, we upsample scenes where the percentage of difficult-to-predict orientations is high. At inference, we let $\hat{\theta}_{ij} = 5° \cdot \arg\max_k \hat{y}_{ij,k}$. With FactoredScenes's pose prediction model, we can generate oriented bounding boxes for each object in given layouts. Notably, our model uses programs as regularization, ensuring that the model generalizes well even when trained on a limited ScanNet dataset. In the Appendix, we include experiments adapting FactoredScenes to infer a pose distribution.

## 3.4 Object Retrieval of ScanNet Objects

Finally, we sample $P(x_1, x_2, \cdots \mid \text{pose}, \text{program})$. Following prior works [24, 27, 21], we retrieve specific 3D instances based on object bounding boxes. Concretely, object retrieval is done by populating the scene with 3D objects whose class and dimensions are the nearest neighbor to the predicted oriented bounding boxes. The objects are then scaled, translated, and rotated to the full predicted poses. To match ScanNet scenes, we manually annotate ScanNet objects with facing directions, such that FactoredScenes's pose model can orient the object accordingly. The set of objects $[x_1, x_2, \ldots]$ form the final scene $S$.

In the object retrieval stage, we process the predicted orientation to set a facing direction, determined by the scene's underlying program. Each object is set to face away from its automatically extracted region boundaries. For primary objects, the region boundaries are taken as those of the full room, containing all objects. For dependent objects, the region boundary is defined as the tight bounding box enclosing the dependency target together with all dependents that share that target (e.g., a cluster of chairs around a table). Notably, this step affects only the facing direction used at retrieval time, and does not change the predicted orientation axis. Overall, our factored design keeps retrieval agnostic to object representation, enabling meshes or point clouds to be flexibly substituted while yielding useful object-centric scenes.

Table 1: Fréchet Inception Distance (FID) and Kernel Inception Distance (KID) comparison of FactoredScenes to prior works and variants of our framework on matching real, ScanNet layouts. The full FactoredScenes framework significantly outperforms all methods.

| | Bedroom FID ↓ | Living FID ↓ | Bedroom KID ↓ | Living KID ↓ |
|---|---|---|---|---|
| DiffuScene [25] | 135.57 | 186.54 | 0.130 | 0.177 |
| Sync2Gen [26] | 126.67 | 139.30 | 0.127 | 0.117 |
| ATISS [24] | 120.27 | 176.95 | 0.117 | 0.166 |
| LayoutGPT [27] | 109.40 | 157.69 | 0.102 | 0.142 |
| FactoredScenes wo/ poses | 101.55 | 137.66 | 0.085 | 0.106 |
| FactoredScenes w/ sampled poses | 110.03 | 123.64 | 0.086 | 0.079 |
| FactoredScenes (Ours) | **67**.**51** | **83**.**49** | **0**.**020** | **0**.**024** |

# 4 Experiments

Our goal is to generate real-world scenes akin to ScanNet, from programs to poses. Here, we evaluate the full FactoredScenes framework in Section 4.1, its program library learning component in Section 4.2, and its pose prediction component in Section 4.3. We discuss limitations and future directions in Section 4.4.

## 4.1 FactoredScenes Evaluation

**Comparison to prior work.** We evaluate FactoredScenes's ability to generate realistic ScanNet-like layouts compared to prior state-of-the-art methods: ATISS [24], DiffuScene [25], Sync2Gen [26], and LayoutGPT [27]. We compare a set of 100 layout images from FactoredScenes and prior work to those of ScanNet. Each layout consists of a set of bounding boxes uniquely colored by their object category. For fairness, we choose a set of intersecting categories between all works and reduce the set of objects in ScanNet accordingly. The final categories are as follows: *armchair, bed, bookshelf, cabinet, chair, coffee table, couch, desk, dresser, lamp, nightstand, shelf, stool, table*. The legend and examples are shown in Figure 5. Following prior work, we evaluate the generated layout images with Fréchet Inception Distance (FID) and Kernel Inception Distance (KID) [39, 40, 41]. Due to limited ScanNet scenes, we highlight KID as a more robust metric.

In Table 1, we see results of FactoredScenes compared to prior works on bedroom and living room scenes. FactoredScenes significantly outperforms all prior works in FID and KID. On bedrooms, our framework shows a 38.3% FID improvement over top prior works, and a 80.4% KID improvement. On living rooms, FactoredScenes yields a 40.1% FID improvement and a 79.5% KID improvement.

**Ablations.** Notably, in Table 1, we study the importance of FactoredScenes's object pose model— by comparing FactoredScenes to a variant without poses (with axis-aligned bounding boxes), and a variant with sampled poses drawn from a normal distribution of orientations specified by class. From the no-pose variant, we see that the layouts created using our learned library already outperform top prior works, as FactoredScenes learns inherent structures within rooms. Importantly, on bedrooms, our sampled-pose variant performs worse than the no-pose variant, highlighting the difficulty of the object pose prediction task; random perturbations fail to capture relationships between objects, and instead yield overlapping boxes with illogical orientations. Overall, the full FactoredScenes framework yields significantly stronger results.

**Human study of real-world scene generation.** To evaluate the quality of full 3D scenes, we conduct a human study via Prolific [42] to compare

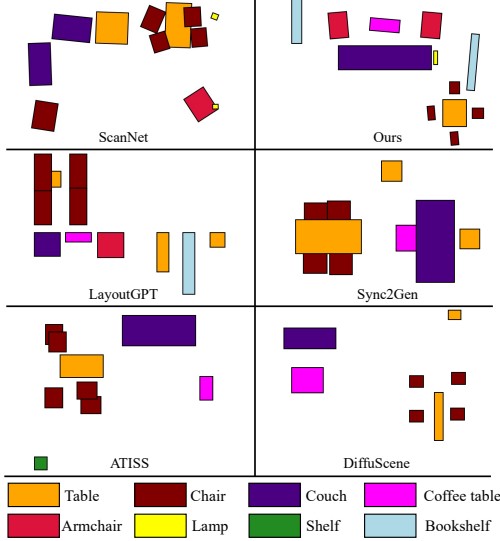

Figure 5: Our predicted layout better matches that of ScanNet scenes compared to prior works.

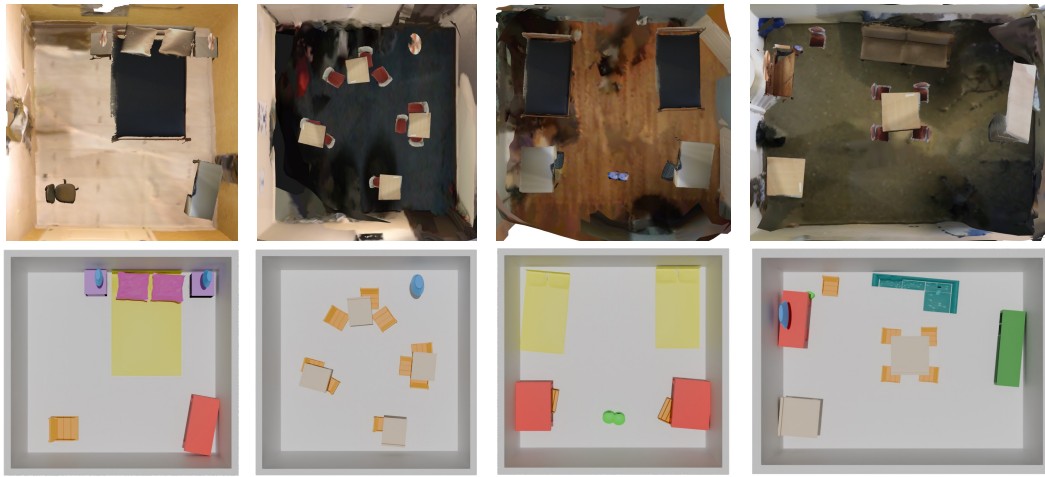

Figure 6: We render diverse examples of FactoredScenes's generated 3D scenes with annotated ScanNet (top row) and ShapeNet (bottom row) objects. Note that ScanNet objects are often partial, hence we include a scaled and interpolated ScanNet background for visualization.

FactoredScenes's generated rooms to ScanNet rooms. We render 3D scenes in a top-down view. Participants were given two rendered rooms, a ScanNet scene and a FactoredScenes scene, and asked: "Which of these two rooms is more realistic and resembles a real-world room?" The questions were randomly ordered and the answer choices shuffled. Out of 400 answers total (20 pairs of scenes and 20 participant answers for each scene), the mean accuracy of choosing the ScanNet scene is **0.67**, indicating that FactoredScenes's generations are difficult to distinguish from that of ScanNet.

**Qualitative examples.** In Figure 6, we present qualitative examples of FactoredScenes's rendered 3D scenes with both ScanNet and ShapeNet objects on the same generated oriented layouts. Our framework is flexible, hence we can easily swap in objects of any type and texture to render scenes. We see that FactoredScenes generates diverse rooms with appropriate structure and varied object poses, all with realistic perturbations (e.g., the orientations of chairs in the second column follows that of the table that they are dependent on). In the Appendix, we provide more visualizations and analyses of rendered 3D scenes.

## 4.2 Learned Program Library Evaluation

Here, we evaluate the diversity of functions used and accuracy of compression with FactoredScenes's learned program library, on both 3D-Front, the dataset it learned from, as well as ScanNet, the target dataset that it generalizes to. We measure diversity as average *high-level* functions used per program, and accuracy of compression as mean intersection over union (mIoU) of reconstruction.

We compare the quality of our learned library to that of an inference-only library in which the LLM proposes the same amount of functions, without seeing examples scenes but given ample context about the task. We parse 100 3D-Front and 150 ScanNet scenes into programs with both libraries, and report results in Table 2. First, we see that our library learned from 3D-Front is able to accurately reconstruct ScanNet scenes, showing that our functions are

Table 2: Comparison of function diversity and compression accuracy between our library and an inference-only library that has never seen example scenes.

|  | 3D-Front | | ScanNet | |
| --- | --- | --- | --- | --- |
|  | Funcs | mIOU | Funcs | mIOU |
| Inf-only library | 1.07 | 0.92 | 0.34 | 0.96 |
| Learned library | **2.89** | **0.96** | **2.53** | **0.98** |

reusable in the context of real scenes, and that there are underlying regularities across human-designed rooms. Second, our learned library that has seen examples scenes significantly outperforms a naive LLM-only approach for 3D-Front and ScanNet, on both diversity and accuracy. Compared to the inference-only library, our learned library shows a 170.1% relative improvement in function use in 3D-Front, and a 644.1% improvement in ScanNet. We note that in ScanNet, lower high-level function use with the inference-only library yields high mIoU, as there is always a trivial solution to

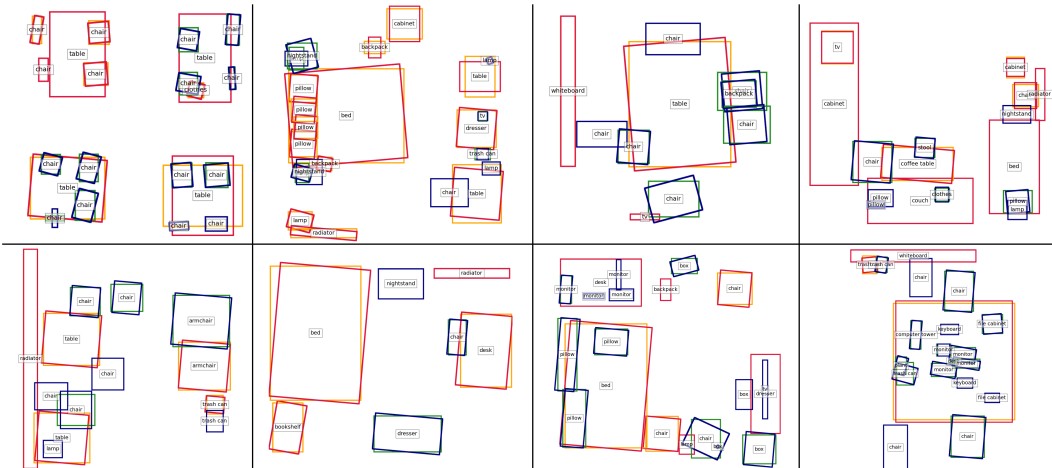

Figure 7: Visualizations of FactoredScenes's pose predictions on the unseen ScanNet test set. Orange and green boxes are the original primary and dependent objects respectively. Red and blue boxes are the predicted primary and dependent objects. Here, we highlight the natural variation in orientations of the primary objects, and the corresponding learned changes in the dependent objects. For example, in the top left scene, the bottom left table is rotated, and as are its dependent chairs correspondingly.

fully reconstruct the input without any compression via functions. Despite this, our library achieves an appropriate trade-off, and significantly outperforms the inference-only library on diversity and accuracy. In the Appendix, we provide examples of ScanNet scenes parsed into compressed programs via our library, and include evaluation of the generative performance from our learned library.

## 4.3 Pose Prediction Evaluation

FactoredScenes's object pose model yields a mIoU of $0.745$ on our unseen ScanNet test set. We visualize test results in Figure 7, and provide more examples and analyses in the Appendix. In Table 3, we report ablation results of our pose prediction model. We compare the model against a variant that does not condition dependent objects' pose predictions on the predictions of their dependent targets. We see that while the

Table 3: Comparison of orientation predictions between FactoredScenes and a variant that does not rely on pose predictions of dependent targets.

|  | Primary $\theta$ acc. | Dep. $\theta$ acc. |
|---|---|---|
| Ours wo/ dep. | 0.537 | 0.397 |
| Ours | **0.542** | **0.442** |

orientation accuracy in degrees of primary objects is similar between the two models, the variant's performance drops significantly for dependent objects.

## 4.4 Discussion

FactoredScenes is framework for real-world scene generation, which decomposes modeling of programmatic room structure and varied object poses. Our method consists of library learning from synthetic data, LLMs to generalize program structure, and programs to regularize pose predictions. FactoredScenes enables learning with limited real-world data to achieve semantically meaningful layouts. However, it is still limited by the LLM's ability to consistently generate valid room programs. Similar to prior LLM-based works, generated scenes occasionally contain unnatural object placements (e.g., aligned nightstands at the middle of the bed, instead of at the head). Additionally, FactoredScenes's pose prediction model is limited by the heuristics used to generate ground-truth oriented bounding boxes for ScanNet. Due to partial objects in ScanNet, extracted orientations are at times inaccurate, yielding illogical overlapping objects. With more accurate labeled data and continued advances in LLMs, we expect FactoredScenes's performance to naturally scale. Its interpretable framework allows each stage to be improved and evaluated independently. In addition, a promising future direction is to include humans in the loop to generate new scenes programs, thus dynamically producing more task-specific data. This high-level scene editing could be achieved with FactoredScenes either via direct program modification or through targeted natural-language prompts.

# 5 Conclusion

Real-world scenes are complex, varied, with limited available data. We propose FactoredScenes as a solution for real-world scene generation, by introducing a factored representation of rooms, decomposed into underlying layout programs and varied object poses. We learn a library of functions on professionally-designed synthetic data, and train a hierarchical object pose model on limited real data. Quantitative experiments and human studies demonstrate that FactoredScenes is a promising step toward synthesizing scenes that are difficult to distinguish from real ones.

## Acknowledgments and Disclosure of Funding

We thank Niladri Shekhar Dutt for providing valuable feedback and visualization guidance. This work is in part supported by the Stanford Institute for Human-Centered AI (HAI), AFOSR YIP FA9550-23-1-0127, ONR N00014-23-1-2355, ONR YIP N00014-24-1-2117, ONR MURI N00014-22-1-2740, and NSF RI #2211258. JH is also supported by the Knight-Hennessy Fellowship and the NSF Graduate Research Fellowship. NM was partially supported by gifts from Adobe Research and the UCL AI Centre.

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

# Supplementary for Factored Real-World
# Scene Generation via Learned Program Libraries

The appendix is organized as the following. In Appendix A, we provide additional experiment results of FactoredScenes across different prompt variations, as well as performance of FactoredScenes on generating a new room type of office. In Appendix B, we present function signatures and descriptions in FactoredScenes's learned library. In Appendix C, we include examples of parsed programs that reconstruct ScanNet scenes. In Appendix D, we report additional predicted orientations on ScanNet and analyze failure cases, as well as add evaluation of adapting FactoredScenes to infer a distribution of poses. In Appendix E, we visualize examples of generated 3D scenes from FactoredScenes. Finally, in Appendix F, we describe details of our model implementation, human study, and broader impact.

## A   Additional Results

Here, we add experiments that test the robustness and sensitivity of our program generation to different prompt variations, under different few-shot prompting configurations (5-shot, 3-shot, 1-shot). In Table 4, FactoredScenes shows consistently strong performance across variants, demonstrating its robustness in generation.

Table 4: FactoredScenes under different few-shot prompting configurations.

| Robustness | All | 5-shot FID | 3-shot FID | 1-shot FID |
|---|---|---|---|---|
| Ours (bedroom) | 64.86 ± 4.36 | 67.51 | 59.83 | 67.25 |
| Ours (living room) | 91.26 ± 6.76 | 83.49 | 95.74 | 94.55 |

While we focus on bedrooms and living rooms in the main text following prior works, we expect the distributions learned from ScanNet to transfer well across many indoor scene types. To showcase performance within indoor domains, we include new results on the office category in Table 5. We follow the same evaluation protocol as in the main text and compute FID and KID between generated layouts and ScanNet office scenes.

Table 5: FactoredScenes generation results on the room category of office.

| | Office FID | Office KID |
|---|---|---|
| Ours | 88.50 | 0.032 |

# B   Program Library

From Figures 8 to 14, we detail the function signatures for all functions in FactoredScenes's learned library.

---

**furniture function**

```python
class furniture:
    def __init__(self, x_min, y_min, x_max, y_max):
        self.x_min = x_min
        self.y_min = y_min
        self.x_max = x_max
        self.y_max = y_max
```

Figure 8: The `furniture` function for primary objects.

---

**parallel function**

```python
def parallel(obj_anchor, distance_apart, direction,
    parallel_object_size=None):
    """
    Place a new furniture object parallel to an existing object based
        on the center point at a specified distance.
    Optionally, specify the size of the new object being placed.

    obj_anchor: reference object to base the new object's position on
    distance_apart: distance between the two objects
    direction: 1 (up), 2 (down), 3 (left), 4 (right)
    parallel_object_size: optional tuple (width, height) to specify the
        size of the new object; defaults to obj_anchor's size
    Returns: a new furniture object positioned parallel to obj_anchor
        with the specified size
    """
```

Figure 9: The `parallel` function for dependent objects.

---

**align function**

```python
def align(obj_ref, count, distance, direction):
    """
    Create a specified number of furniture objects aligned in a given
        direction
    with a specified distance between them, based on a single reference
        object.

    obj_ref: reference furniture object to be aligned
    count: number of objects to instantiate and align
    distance: distance between consecutive objects
    direction: 1 (up), 2 (down), 3 (left), 4 (right)
    Returns: a list of aligned furniture objects
    """
```

Figure 10: The `align` function for dependent objects.

```python
def grid(obj_ref, rows, cols, h_distance, v_distance):
    """
    Create a grid of furniture objects based on a single reference
        object, with respect to its center.
    The grid has specified rows and columns, with horizontal and
        vertical distances between objects.
    The grid grows in both directions relative to the center of the
        obj_ref.

    obj_ref: reference furniture object to generate the grid
    rows: number of rows in the grid
    cols: number of columns in the grid
    h_distance: horizontal distance between objects in the grid
    v_distance: vertical distance between objects in the grid
    Returns: a list of furniture objects arranged in a grid
    """
```

Figure 11: The `grid` function for dependent objects.

```python
def grid_with_offset(obj_ref, rows, cols, h_distance, v_distance,
    row_offsets=None, col_offsets=None):
    """
    Create a grid of furniture objects with optional row and column
        offsets,
    with respect to the center of the reference object (obj_ref).

    obj_ref: reference object to determine the grid's location and size
    rows: number of rows in the grid
    cols: number of columns in the grid
    h_distance: horizontal distance between objects in the grid
    v_distance: vertical distance between objects in the grid
    row_offsets: list of offsets for specific rows (optional)
    col_offsets: list of offsets for specific columns (optional)
    Returns: a list of furniture objects arranged in a grid with
        offsets relative to obj_ref's center
    """
```

Figure 12: The `grid_with_offset` function for dep. objects.

```python
def symmetrical(center, distance_x, distance_y,
    symmetrical_objects_size):
    """
    Place four objects symmetrically around a central point, based on
        the specified object size.
    The placement is based on the center of each symmetrical object.

    center: (x, y) coordinates of the central point
    distance_x: horizontal distance from the center to the new objects
    distance_y: vertical distance from the center to the new objects
    symmetrical_objects_size: tuple (width, height) specifying the size
        of the symmetrical objects
    Returns: a list of four furniture objects symmetrically placed
        around the center
    """
```

Figure 13: The `symmetrical` function for dependent objects.

```
cluster_placement function

def cluster_placement(obj_center, offsets, clustered_objects_size=None)
    :
    """
    Place a cluster of furniture objects around a central object based
        on specified offsets.
    The offsets are with respect the the obj_center center coordinates.
    Optionally, specify the size of the clustered objects.

    obj_center: central furniture object used as the anchor point
    offsets: list of (x_offset, y_offset) tuples for placing
        surrounding objects
    clustered_objects_size: optional tuple (width, height) to specify
        the size of the clustered objects;
                            defaults to the size of the central object
    Returns: a list of furniture objects placed around the central
        object
    """
```

Figure 14: The `cluster_placement` function for dep. objects.

# C  ScanNet Program Parsing

In Figure 15, we include examples of parsed programs that reconstruct ScanNet scenes. We highlight the apt uses of high-level functions, for instance, in the top example, the `parallel` functions for *beds*, *desks*, *dressers*, etc. In our experiments, we note that GPT-o1 is exceedingly proficient at parsing input bounding boxes in text form into programs with our learned library, and has strong potential for extracting program-based representation from structured data.

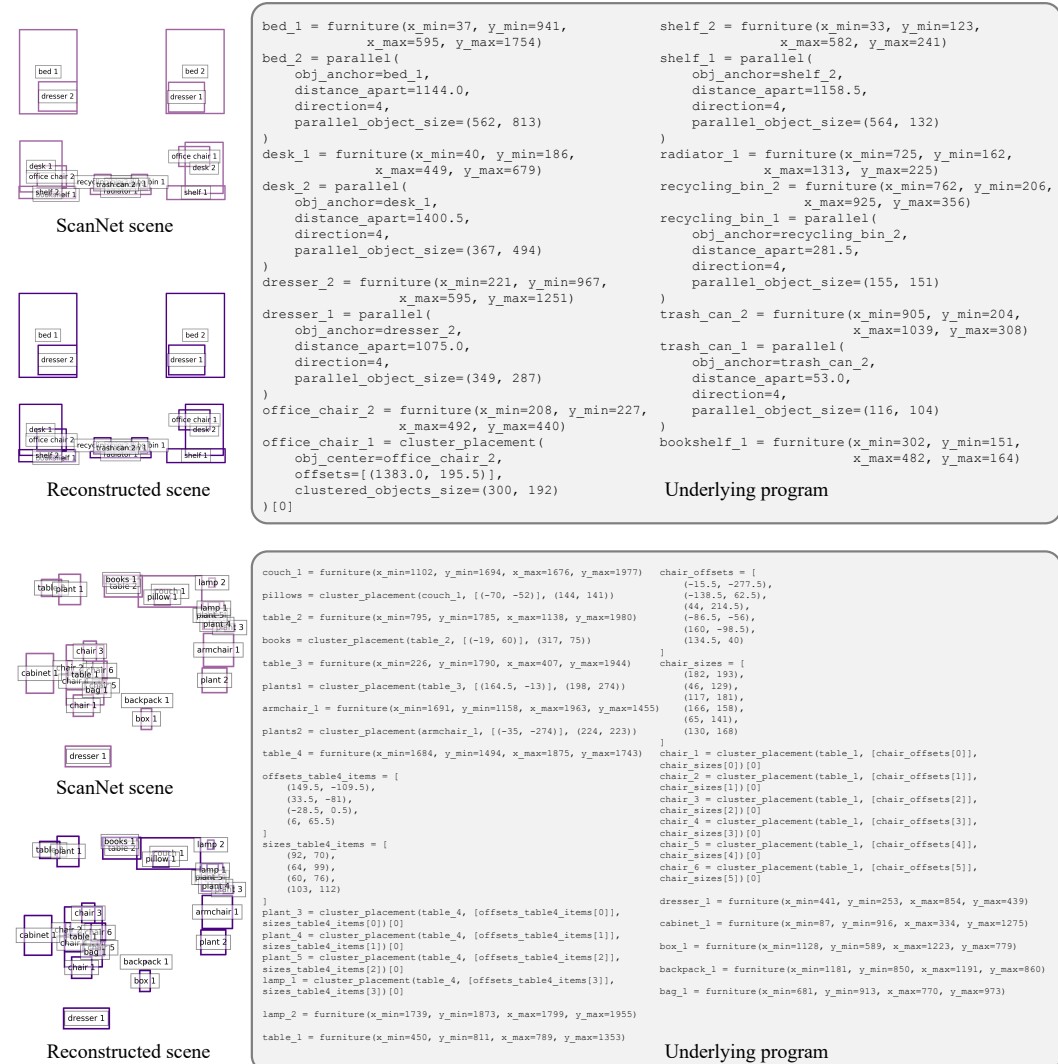

Figure 15: Examples of programs underlying complex ScanNet scenes, parsed with LLMs using FactoredScenes's learned library.

# D   Predicted Object Poses

In Figure 16, we present additional examples of FactoredScenes's predicted object poses on an unseen ScanNet test set. We highlight several failure cases in the last row. In particular, the left two scenes in the bottom row have infeasible rotations (e.g., *kitchen cabinet* and *sink*), while the right two scenes in the bottom row have no dependent objects, leading to less logically dependent poses. We believe that a combination of more accurately labeled data and a larger quantity of data would improve FactoredScenes's object pose model significantly.

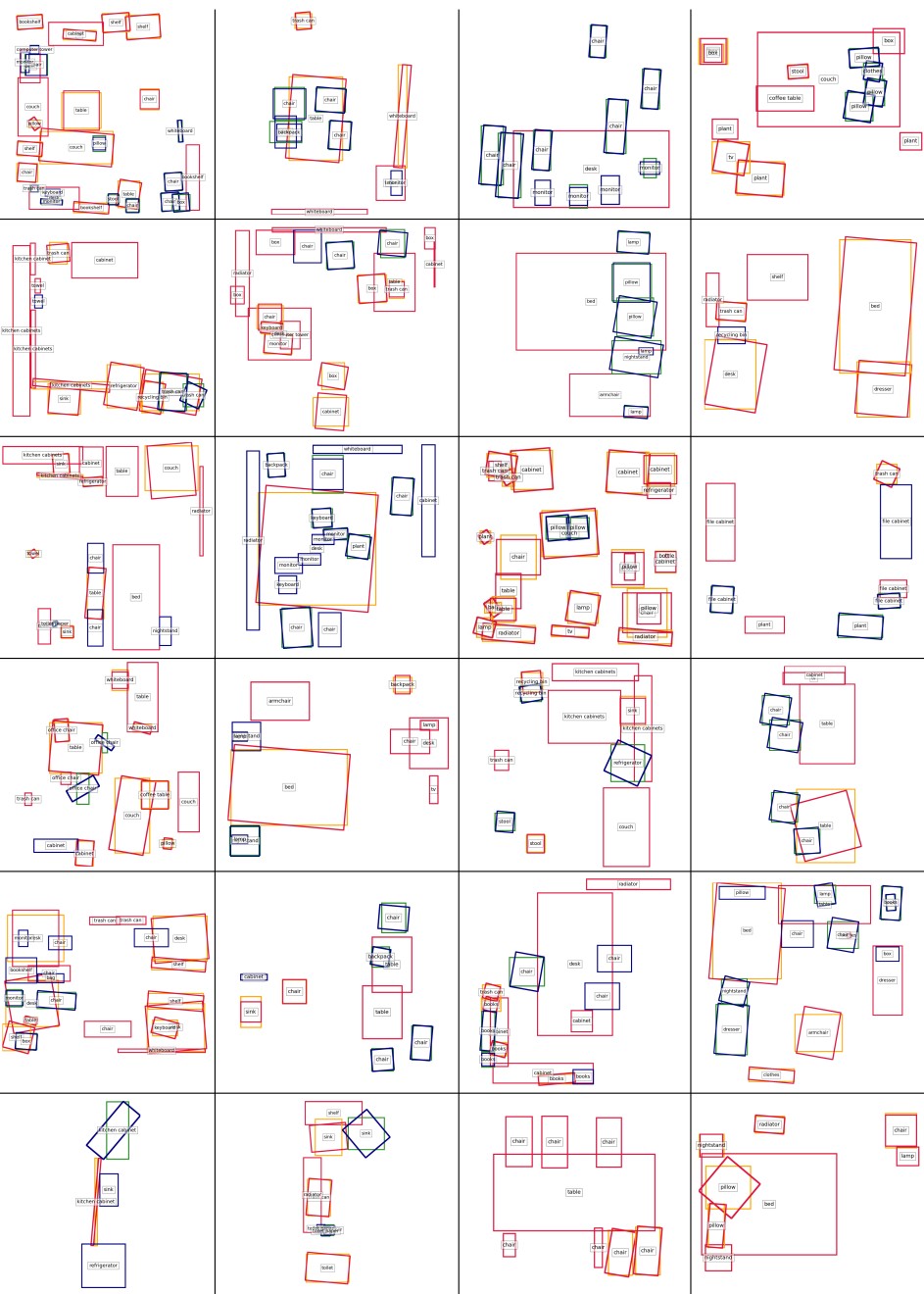

Figure 16: Additional visualizations of FactoredScenes's pose predictions on the unseen ScanNet test set. Orange and green boxes are the original primary and dependent objects respectively. Red and blue boxes are the predicted primary and dependent objects.

We additionally report results of adapting FactoredScenes to infer a distribution of poses rather than a single pose for a given layout, trained with negative log likelihood (NLL). In Table 6, we report NLL on the ScanNet test set, and see that our trained model significantly improves on both a random (untrained) baseline and an informed fixed mean and variance baseline. In Table 7, we report a point estimate metric by taking the mean of the predicted distribution. Our generative model outperforms all top prior works (Sync2Gen for living rooms and LayoutGPT for bedrooms), though less than our predictive model. We note that in this framework, while the means and standard deviations of dependent objects are conditioned on that of its dependency target, the sampling is conducted independently. Hence, it is unable to robustly align dependent objects' orientations correspondingly.

Table 6: NLL comparison of our pose model to baselines.

|  | ScanNet Test NLL ↓ |
| --- | --- |
| Untrained model baseline | $1.795 \times 10^{18}$ |
| Fixed mean & variance baseline | $2.279 \times 10^{6}$ |
| Generative FactoredScenes | $\mathbf{1.587 \times 10^{1}}$ |

Table 7: Comparison using mean-based point estimates.

|  | Living room FID ↓ | Bedroom FID ↓ | Living room KID ↓ | Bedroom KID ↓ |
| --- | --- | --- | --- | --- |
| Top prior | 139.30 | 109.40 | 0.117 | 0.102 |
| Generative | **113.06** | **99.95** | **0.067** | **0.071** |

# E Generated Scenes

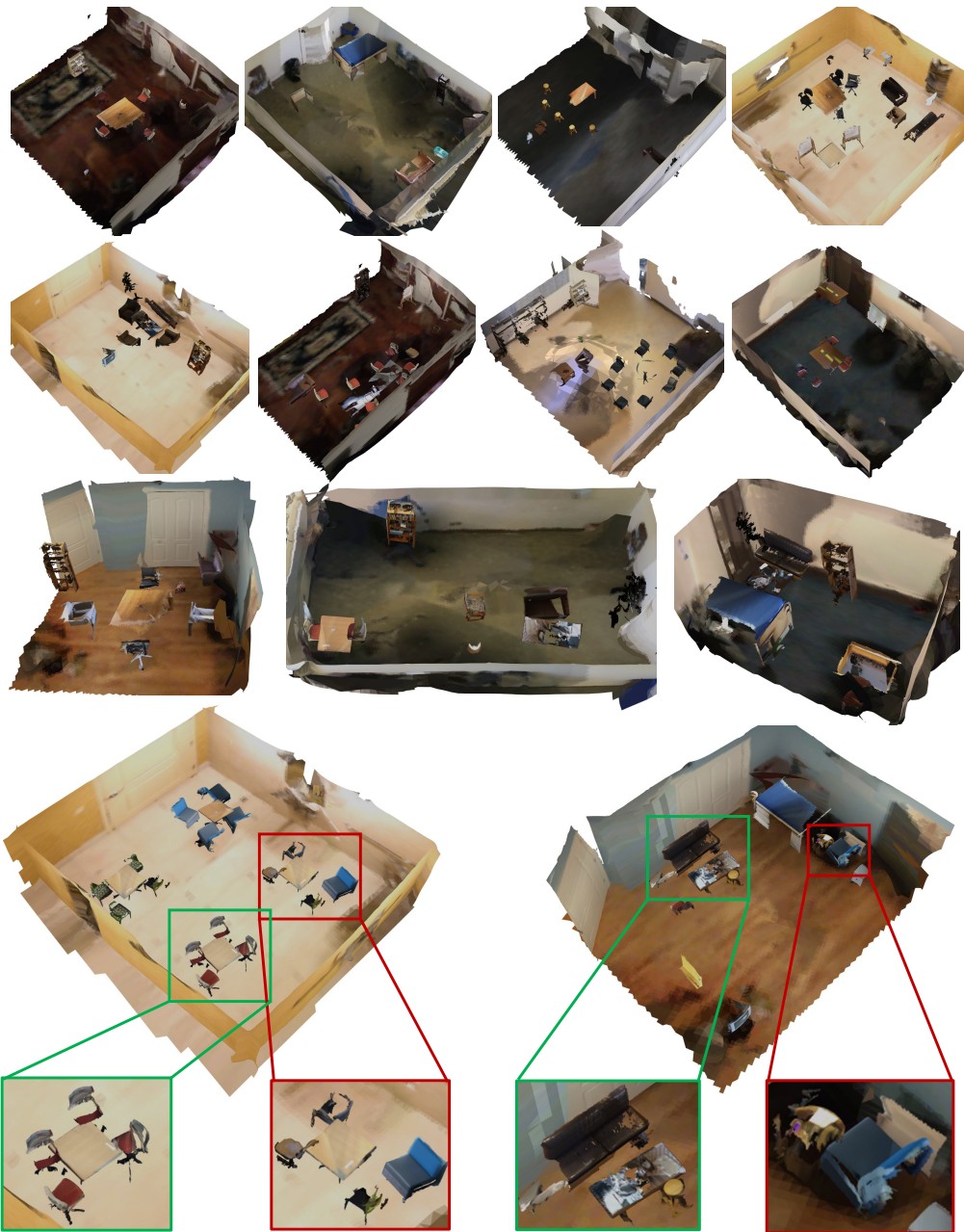

Figure 17: Additional examples of FactoredScenes's generated rooms.

We present additional examples and analyses of FactoredScenes's generations in Figure 17, as well as showcase specific success and failure cases in the bottom row. In the green box of the left scene, we see that our framework correctly generates chairs of the same type oriented towards the table. Notably, this is without a priori specifying any constraints around how chairs should be oriented around tables or that chairs around a table are usually of the same type. The predicted orientations and facing directions are natural. However, in the red box, we see an example where not only are chairs of different types (as they are not accurately parsed in the program as objects dependent on the table), but are also faced in unusual ways (for a similar reason, as the region boundaries here are the room walls).

Similarly, in the green box of the right scene, we see a very natural setup, of a couch facing a coffee table and a stool across from it, with all objects oriented appropriately. But in the red box, we see a chair facing towards the bed in an unrealistic pose.

Though FactoredScenes can accurately model detailed parts of scenes, such as specific orientations of chairs, it occasionally errors in (i) generating valid programs with correct dependent objects and relations, and (ii) predicting reasonable object poses when objects are in unfamiliar positions. We believe these aspects can be improved with better-performing LLMs as well as human-annotated data for orientations instead of heuristics-based labels.

# F   Details

**Model training.**   Here, we describe the FactoredScenes settings. The library learning component uses the 3D-Front [2] dataset, released under the CC BY-NC-SA 4.0 license; and the orientation model is trained on the full ScanNet dataset [1], released under the MIT license. When training our pose prediction model, we used $581$ scenes with accurately parsed programs out of $707$ total unique scenes; our train set comprised $523$ scenes and test set comprised $58$ scenes. We used the Adam optimizer with a learning rate of $0.0001$, and trained our models on a single Titan RTX GPU with $24$ GB of memory.

**Human experiments.**   We conducted a human study via Prolific [42] to compare FactoredScenes's generated rooms to ScanNet rooms, with the following instruction: "In this study, you will be asked questions comparing two images of rooms: which of these two rooms is more realistic and resembles a real-world room?". The questions were randomly ordered and the answer choices shuffled. We queried 20 participants over 20 pairs of scenes each, with an average compensation per participant of 25 USD per hour. We identified no potential risks for the study, as each image shown is a rendering of a 3D scene, and did not seek IRB approval, as the task involved no personally identifiable information or sensitive content.

**Broader impacts.**   Our work focuses on generating realistic scenes, for the purpose of creating real-world, object-centric datasets that the community can build upon. While there are no direct routes to harm with our model, we acknowledge its potential misuse, such as in pipelines for generating fake content. In addition, although we use a large language model to produce layout programs regularized by our learned library, our framework remains susceptible to biases inherent in such models. We encourage future work to explore bias mitigation strategies when deploying these systems.

