# OpenReview forum: "From Programs to Poses: Factored Real-World Scene Generation via Learned Program Libraries"
_NeurIPS.cc/2025/Conference — NeurIPS 2025 poster_

### Official Review · Reviewer_e9Et · 2025-07-01

**Clarity:** 2
**Significance:** 2
**Originality:** 2
**Rating:** 4
**Confidence:** 3

**Summary:**

FactoredScenes presents a modular framework for 3D scene generation that combines layout programs with learned object pose prediction. A function library is first learned from 3D-Front data. A neural model then predicts object poses conditioned on the program structure, and real 3D assets are retrieved and placed accordingly. The method achieves high-quality, diverse scene synthesis compared to previous methods.

**Questions:**

Can you elaborate on the model details, particularly the architecture of the pose prediction network? how the model architecture was chosen and ablated? Adding illustrations or diagrams to clarify the model architecture and technical components might further help.

Could you provide more analysis on the system’s robustness and any other failure modes? Specifically, how sensitive is the pipeline to errors in program generation, and how do such upstream errors affect downstream pose prediction and object retrieval quality?

**Ethical Concerns:**

["NO or VERY MINOR ethics concerns only"]

**Final Justification:**

After reading the rebuttal, I'd like to raise my score.

**Limitations:**

While the use of LLMs for layout program generation is effective, it is not particularly novel and the presented results are still of relative low visual fidelity. Following the line of LayoutGPT, Holodeck, and LayoutVLM, the proposed tailored mechanism library learning contributes to automating the extraction of a relation library on the code-level. However, the demonstrated improvements with more relation descriptions and noisy object pose remain relatively incremental. Compared to the current code-based approach, VLM-based extraction appears more promising for directly capturing relational patterns from real-world examples in the long-term. It may be interesting to incoporate more advanced explorations, such as expanding ground-plane layouts only to full 3D environments with vertical arrangements, or multi-level structures with hierarchical dependencies. Physical plausibility is not explicitly modeled or enforced, which may be considered to be added in LLM program generation. Including discussion or experiments on these fronts would strengthen the paper.

**Quality:**

3

**Strengths And Weaknesses:**

Overall it proposed a relative comprehensive solution with well-engineered, modular pipeline with clear stage-wise design. The design is well-motivated and the leverage of LLM and neural model for pose prediction is quite reasonable.

The final rendered visuals exhibit relatively low fidelity. The two datasets used (3D-Front and ScanNet) are relatively limited in diversity and scale. Although the LLM-based program generation is a central component, it is treated largely as a black box, with little discussion of prompt sensitivity or the validity rate of generated programs. Some model details are under-specified, such as the pose predictor, which is only described as “attention-based.”

---

> ### Author Rebuttal · Authors · 2025-07-30
>
> Thank you for your constructive feedback, which has greatly improved the quality of our paper.
>
> **Q: Additional analyses on robustness and failure modes of program generation.**
>
> Thanks for the suggestion. We agree that it is valuable to analyze the capabilities of FactoredScenes’ LLM-based program generation. Here, we add experiments that test the validity of FactoredScenes’ generated programs, as well as the robustness of downstream results to potential errors in these programs.
>
> First, we report the percentage of FactoredScenes’s generated programs that execute successfully without errors and the percentage of programs that are semantically valid. As there is no ground truth for semantically valid programs, we define semantic validity using a conservative heuristic: a generated layout is deemed invalid if any pair of objects (excluding lamps and pillows) has IoU > 0.3. Here, we see that a high percentage of programs execute successfully, and even with this conservative, semantically valid criteria, we see a high percentage of programs marked as valid, which indicates that catastrophic errors are rare.
>
> | Across 100 rooms | % of program that executes successfully | % of programs that are semantically valid |
> |------------------|-----------------------|-----------|
> | Ours             | 97%                      | 90%             |
>
> Second, we evaluate the effect of FactoredScenes’ program validity on downstream generation results, and report performance for scenes generated from valid vs. invalid programs. As shown below, semantically valid programs as per our heuristic produce lower (better) performance, although the difference is slight.
>
> | Generation | KID of semantically **valid** programs | KID of semantically **invalid** programs |
> |-----------|------------|----------|
> | Ours       | 0.0183     | 0.0192  |
>
> Lastly, we investigate the robustness and sensitivity of our program generation to different prompt variations, under different few-shot prompting configurations (5-shot, 3-shot, 1-shot). FactoredScenes shows consistently strong performance across variants, significantly above that of prior works, demonstrating its robustness in generation.
>
> | Robustness        | All           | 5-shot FID | 3-shot FID | 1-shot FID |
> |-------------------|---------------|------------|------------|------------|
> | Ours bedroom      | 64.86 ± 4.36 | 67.51      | 59.83      | 67.25      |
> | Ours living room  | 91.26 ± 6.76 | 83.49      | 95.74      | 94.55      |
>
> We also provide the prompts used for both library learning and program generation below, and will include this as well as expanded discussions in the revised paper.
>
> Library learning:
>
> *You have the following functions: [Current set of functions in the library]. Can you parse the below bounding boxes into programs with the above functions, such that they can be accurately reconstructed? The bounding boxes are defined by name and corresponding coordinates x_min, y_min, x_max, y_max representing the bottom left and top right corners of the bounding box. Please parse the bounding boxes into programs such that we only need to define the coordinates of minimal furniture only. The rest of the objects should be defined as so without coordinates and named (e.g., object_x = function(...)). Return code to generate the objects in a dictionary named objects, where the key is the original name of the object, and the value is the newly created object by the program. Bounding boxes: [Input bounding boxes].*
>
> *The following are programs describing layouts with bounding boxes, each program separated by ---. The current functions used are as follows: [Current set of functions in the library]. Can you propose new functions to compress these programs while retaining the same bounding box reconstructions? Compress programs by minimizing the number of objects that need explicit coordinate definition. Please generate the full definition and code for these functions.*
>
> Program generation:
>
> *You have the following functions: [Full set of functions in the library]. These functions generate bounding boxes representing rooms. The bounding boxes are defined by name and corresponding coordinates x_min, y_min, x_max, y_max representing the bottom left and top right corners of the bounding box. Below are programs that generate a [Room type]. Note that each object is named based on its semantic class: [Few-shot examples]. Now, please generate another [Room type] using the given functions. You can generate this room by using different functions, locations, and object classes, but make sure they are reasonable objects to be in the room. Think step-by-step about where objects should be placed and what their sizes are, such that they are reasonable within the example program. Still return a dictionary called objects, do not repeat the given function implementations. Return code only. Only use the following object classes: [Objects categories from ScanNet].*
>
> **Q: Comparison to prior LLM-based works and the importance of pose predictions.**
>
> We highlight that, unlike prior LLM-based methods (e.g., LayoutGPT, Holodeck, LayoutVLM) that operate with fixed, hand-crafted libraries, FactoredScenes learns a library of structural abstractions directly from data, and constrains LLM generation using this learned library. We hypothesize that this learnt structural prior helps reduce unnatural generations by anchoring the generation space to abstractions learned from example rooms, which leads to the strong performance of the FactoredScenes wo/ poses variant in Table 1.
>
> Additionally, a key contribution of FactoredScenes is that it enables training on diverse datasets with noisy object poses, such as ScanNet (and hence generate real-world, lived-in scenes with object layouts), while prior works cannot, because they are limited to learning from axis-aligned objects from datasets like 3D-Front. We propose FactoredScenes as a structured framework precisely to enable learning from diverse data sources, in a data-efficient way with appropriate underlying models, such that we can generate realistic scene layouts.
>
> **Q: Clarification of model details.**
>
> Our pose prediction model predicts object poses in a hierarchical manner using per-object MLP encoders and self-attention modules, shown in Figure 4. For each object, we concatenate features including: category label, bounding box coordinates, closest wall encoding, and global wall features. These are passed through a Transformer-style self-attention block. Specifically, we apply layer normalization followed by learned linear projections to obtain queries, keys, and values, which are then processed via a multi-head attention module. The attention output is added back through a residual connection and passed through a feedforward network (two-layer MLP with ReLU) to predict pose. For dependent objects, we additionally encode the dependency function and the orientation of the source object. We include an ablation of the hierarchical design in Table 3, and will clarify these architectural details in the main text and add additional figures to illustrate the specific technical components.
>
> **Q: Limitations of 3D-Front and ScanNet.**
>
> While most prior works only train on 3D-Front, FactoredScenes additionally enables training on ScanNet, which allows learning of object poses in real-world, lived-in scenes. We agree that the limiting aspect here is the availability of real-world datasets, which we will discuss in depth in our limitations section. However, we highlight that while small, ScanNet covers a wide range of indoor scene categories: apartment, living room, lounge, kitchen, hallway, bedroom, hotel, lobby, bathroom, office, classroom, gym, mail room, bookstore, library, stairs, mailboxes, computer cluster, storage, basement, garage, conference room, closet, dining room, copy room, game room, laundromat, laundry room.
>
> Hence, we expect the distributions learned from ScanNet to transfer well across many indoor scene types. To showcase performance within indoor domains, we include new results on the office category. We follow the same evaluation protocol as in the main paper and compute FID and KID between generated layouts and ScanNet office scenes.
>
> |      | Office FID | Office KID |
> |------|------------|------------|
> | Ours | 88.50    | 0.032     |
>
> **Q: Quality of rendered visuals.**
>
> We clarify that the final rendered visuals use objects from ShapeNet and ScanNet, which are often partial or low-resolution. However, FactoredScenes’s generated layout enables us to flexibly substitute higher-fidelity meshes or point clouds during retrieval to improve visual quality without changing the generation pipeline. We will clarify and aim to improve visualizations in the revised paper.
>
> **Q: Discussions on VLM-based extraction, vertical arrangements, multi-level structures, and physical plausibility.**
>
> We thank the reviewers for these suggestions, and agree that incorporating VLM-based extraction is a promising direction, especially for unlabeled scenes. We note that FactoredScenes currently captures relational patterns from real-world data via annotated object layouts in ScanNet. In the absence of such ground truth object poses, VLM-based extraction would be a valuable approach, and we view this as an exciting direction for future work.
>
> We also agree that expanding to vertical arrangements, multi-level structures, and physical plausibility constraints is exciting and would augment the generation process. Currently, we focus on ground-plane layouts with a two-level hierarchy due to limited ScanNet data. However, the modular nature of FactoredScenes allows for incorporating more geometric reasoning and physical constraints (or other text-conditioned constraints) into future generations as data and models improve. Thank you for the feedback, and we will add discussions on these directions in the revised paper.

---

> > ### Comment · Reviewer_e9Et · 2025-08-06
> >
> > Thanks for the clarification and additional details. The rebuttal has partially addresses my major concerns and I'd like to raise my rating.

---

> > > ### Author Response · Authors · 2025-08-07
> > >
> > > Thank you again for the constructive comments. We will be sure to incorporate all clarifications and details into the revised paper, and we appreciate your feedback.

---

### Official Review · Reviewer_uVTa · 2025-07-02

**Clarity:** 2
**Significance:** 2
**Originality:** 3
**Rating:** 4
**Confidence:** 5

**Summary:**

This paper introduces FactoredScenes, a framework for real-world unconditional scene synthesis with lived-in scenes, by decomposing the scene into programs encoding the layout and object poses capturing real-world variation.

First, FactoredScenes learns a library of functions from a synthetic dataset through an iterative wake-sleep formulation using LLMs. Then, it generates new programs using LLMs conditioned on the learned library and parsed program samples from a real-world dataset. A neural network learns predicting the exact object poses hierarchically and finally the framework retrieves the object instances producing the 3D scene.

**Questions:**

All of my questions are listed in the weaknesses section, and I may adjust the rating if they are well addressed.

**Ethical Concerns:**

["NO or VERY MINOR ethics concerns only"]

**Final Justification:**

Thank you for responding to my questions, the rebuttal addresses most of my concerns. I will keep my rating positive.

**Limitations:**

Yes.

**Quality:**

3

**Strengths And Weaknesses:**

Strengths:

1.	Real-world unconditional scene synthesis with lived-in scenes is an interesting problem, as existing methods mostly employ synthetic datasets and attempt to produce similar scenes.
2.	Learning a library of reusable functions to capture structural relationships and generating scene-level programs based on this learned library is novel.
3.	Decomposing scene generation in several steps and hierarchical pose prediction based on object dependencies is intuitive.

Weaknesses:

1)	The authors claim their method generalizes to new scene types. However, all experiments are conducted only on the ScanNet dataset, which is also used to train the object pose estimation network. This raises concerns about dataset-specific biases in pose distributions and scene layouts that may not apply to other real-world indoor scenes. Further evaluations on different real-world datasets are required to support this claim.
2)	Lines 168–169: The verification and correction mechanism is not clear. Is there a feedback loop that runs until the predicted programs accurately reconstruct the input bounding boxes for a given scene?
3)	The success of library learning (Section 3.1) and program generation (Section 3.2) relies heavily on LLM prompting. However, the paper does not specify how to interact with the LLM during library learning or how these instructions are structured.
4)	The paper performs the final evaluation on ScanNet. However, the baseline methods, whether using in-context learning or fully learning-based approaches, have only seen synthetic datasets and cannot capture real-world variations. To fairly evaluate scene-level program generation against these baselines, layouts should either also be generated and evaluated on 3D-FRONT (eliminating the real-world component) or the proposed pretrained pose estimation model should be applied to the layouts produced by the baseline models to produce ScanNet like scenes.

---

> ### Author Rebuttal · Authors · 2025-07-30
>
> Thank you for your constructive feedback, which has greatly improved the quality of our paper.
>
> **Q: Additional evaluations on 3D-Front without the real-world component.**
>
> We highlight that one of the main contributions of FactoredScenes is precisely that it enables training across diverse datasets, such as ScanNet (and hence generate realistic, lived-in scenes with object layouts), while prior works cannot, because they are limited to learning from axis-aligned layouts from datasets like 3D-Front. While not the focus of our paper, we add additional evaluations of FactoredScenes and prior works on 3D-Front, removing the real-world component. We see that FactoredScenes outperforms prior works on bedrooms, while achieving competitive performance to prior works on living rooms, despite not being designed to optimize for this task. Thank you for your suggestion, and we will include this evaluation in the revised paper and clarify our focus.
>
> | 3D-Front | Bedroom KID | Living KID |
> |--------------------|-------------|------------|
> | DiffuScene         | 0.036      | 0.088     |
> | Sync2Gen           | 0.010      | **0.030**     |
> | ATISS              | 0.052      | 0.093     |
> | LayoutGPT          | 0.030      | 0.038     |
> | Ours               | **0.009**      | 0.031     |
>
> **Q: Prompts to LLMs and their robustness.**
>
> We provide the prompts used for both library learning and program generation below, and will include this in the revised paper.
>
> Library learning:
>
> *You have the following functions: [Current set of functions in the library]. Can you parse the below bounding boxes into programs with the above functions, such that they can be accurately reconstructed? The bounding boxes are defined by name and corresponding coordinates x_min, y_min, x_max, y_max representing the bottom left and top right corners of the bounding box. Please parse the bounding boxes into programs such that we only need to define the coordinates of minimal furniture only. The rest of the objects should be defined as so without coordinates and named (e.g., object_x = function(...)). Return code to generate the objects in a dictionary named objects, where the key is the original name of the object, and the value is the newly created object by the program. Bounding boxes: [Input bounding boxes].*
>
> *The following are programs describing layouts with bounding boxes, each program separated by ---. The current functions used are as follows: [Current set of functions in the library]. Can you propose new functions to compress these programs while retaining the same bounding box reconstructions? Compress programs by minimizing the number of objects that need explicit coordinate definition. Please generate the full definition and code for these functions.*
>
> Program generation:
>
> *You have the following functions: [Full set of functions in the library]. These functions generate bounding boxes representing rooms. The bounding boxes are defined by name and corresponding coordinates x_min, y_min, x_max, y_max representing the bottom left and top right corners of the bounding box. Below are programs that generate a [Room type]. Note that each object is named based on its semantic class: [Few-shot examples]. Now, please generate another [Room type] using the given functions. You can generate this room by using different functions, locations, and object classes, but make sure they are reasonable objects to be in the room. Think step-by-step about where objects should be placed and what their sizes are, such that they are reasonable within the example program. Still return a dictionary called objects, do not repeat the given function implementations. Return code only. Only use the following object classes: [Objects categories from ScanNet].*
>
> In addition, we add experiments to investigate the sensitivity of our program generation to different prompt variations, under different few-shot prompting configurations (5-shot, 3-shot, 1-shot). We see that FactoredScenes shows consistently strong performance across variants, significantly above that of prior works, demonstrating its robustness in generation.
>
> | Robustness        | All           | 5-shot FID | 3-shot FID | 1-shot FID |
> |-------------------|---------------|------------|------------|------------|
> | Ours bedroom      | 64.86 ± 4.36 | 67.51      | 59.83      | 67.25      |
> | Ours living room  | 91.26 ± 6.76 | 83.49      | 95.74      | 94.55      |
>
> **Q: Generalization to other real-world indoor scenes.**
>
> Thank you for raising this point. We clarify that FactoredScenes aims to generalize in two key ways: (1) our LLM-based program generator does not rely on handcrafted & fixed libraries and can generate novel scene layouts beyond ScanNet (as well as learn new libraries as needed), and (2) our pose model is evaluated on new LLM-generated rooms not seen during training, demonstrating generalization beyond the training ScanNet scenes. We show in the main text that our pose prediction model can accurately generate poses for such new indoor rooms.
>
> However, the pose model is trained on ScanNet object orientations, so generalization to domains with fundamentally different pose distributions (e.g., outdoor or industrial scenes) would require new pose data, though we expect the distributions learned from ScanNet to transfer well across many indoor scene types. To showcase performance within indoor domains, we include new results on the office category, which did not require newly handcrafted function definitions in order to generate. We follow the same evaluation protocol as in the main paper and compute FID and KID between generated layouts and ScanNet office scenes.
>
> |      | Office FID | Office KID |
> |------|------------|------------|
> | Ours | 88.50    | 0.032     |
>
> We agree that the limiting aspect here is the availability of real-world datasets, which we will highlight in our limitations section. Additionally, we note that while ScanNet is small, it spans diverse real-world indoor scene categories: apartment, living room, lounge, kitchen, hallway, bedroom, hotel, lobby, bathroom, office, classroom, gym, mail room, bookstore, library, stairs, mailboxes, computer cluster, storage, basement, garage, conference room, closet, dining room, copy room, game room, laundromat, laundry room. We will expand on this discussion in the revised paper.
>
> **Q: Verification and correction mechanism.**
>
> Indeed, in the library learning stage, FactoredScenes uses a reconstruction-based verification loop. If the program-generated layout differs from the ground truth input by more than a distance threshold, it is considered incorrect. We allow the LLM up to 16 attempts per scene to refine the program until a successful reconstruction is achieved. Thank you for your feedback, and we will clarify this in the main text.

---

> > ### Comment · Reviewer_uVTa · 2025-08-05
> > **Official Comment by Reviewer uVTa**
> >
> > Thank you for responding to my questions, the rebuttal addresses most of my concerns. I will keep my rating positive.

---

> > > ### Author Response · Authors · 2025-08-05
> > >
> > > Thank you again for the constructive feedback. We will incorporate all discussions and added evaluations into our revised paper.

---

### Official Review · Reviewer_ziVh · 2025-07-02

**Clarity:** 3
**Significance:** 3
**Originality:** 3
**Rating:** 4
**Confidence:** 4

**Summary:**

This paper focuses on realistic 3D scene generation with varied object layouts and poses. A new framework, i.e., FactoredScenes, is proposed by leveraging the underlying room structure. A factored room representation is proposed to decompose scenes into hierarchically organized concepts of programs and object poses in lived-in scenes. The experimental results show the effectiveness of the proposed method for realistic room generation.

**Questions:**

1. Based on the results in Table 1, the results of the no-pose variant already outperform existing works. Why the proposed method can learn inherent structures better than existing works?
2. Is it possible to be generalizable to other scenarios other than lived-in scenes?

**Ethical Concerns:**

["NO or VERY MINOR ethics concerns only"]

**Final Justification:**

I appreciate the authors' clarifications and additional comparisons. My main concerns have been addressed by the rebuttal. Although the proposed method is slower than baselines due to structured program execution and LLM querying, it still provides a new way to synthesize realistic 3D scenes with the discovery of reusable structural abstractions underlying scenes.

I would lean to accept the paper by involving the additional evaluations and discussions in the revised version.

**Limitations:**

Yes.

**Paper Formatting Concerns:**

No.

**Quality:**

3

**Strengths And Weaknesses:**

The idea of decomposing the room into hierarchical concepts is interesting, making it possible to leverage different data sources for realistic scene generation. Semantic knowledge from LLMs is further distilled to capture structural relationships between objects in rooms. The results demonstrate the effectiveness of the proposed method.

The main weaknesses are listed as follows.
1. The evaluations are not thorough. The proposed method consists of five steps for scene generation. The intermediate results from each step should be evaluated and discussed in the experiments.
2. GPT-o1 is used in the current experiment, but the importance of GPT-o1 in the whole pipeline has not been studied yet. The ablation studies do not consider the effect of different LLM settings, including the parameters of GPT-o1 and replacing GPT-o1 with other methods.
3. A detailed computational cost and running time comparison between the proposed method and existing works is necessary.

---

> ### Author Rebuttal · Authors · 2025-07-30
>
> Thank you for your constructive feedback, which has greatly improved the quality of our paper.
>
> **Q: Effect of different LLM settings.**
>
> We agree that it is valuable to explore the effect of different LLM settings. Based on your suggestion, we add experiments that test the robustness and sensitivity of our program generation to different prompt variations, under different few-shot prompting configurations (5-shot, 3-shot, 1-shot). FactoredScenes shows consistently strong performance across variants, significantly above that of prior works, demonstrating its robustness in generation.
>
> | Robustness        | All           | 5-shot FID | 3-shot FID | 1-shot FID |
> |-------------------|---------------|------------|------------|------------|
> | Ours bedroom      | 64.86 ± 4.36 | 67.51      | 59.83      | 67.25      |
> | Ours living room  | 91.26 ± 6.76 | 83.49      | 95.74      | 94.55      |
>
> **Q: Runtime comparisons.**
>
> Thank you for the suggestion. We have now added a runtime and API usage comparison across methods, averaged over 10 generations on a single Titan RTX GPU. We see that FactoredScenes is slower than fully neural baselines like ATISS and Sync2Gen due to structured program execution and LLM querying, but remains comparable to DiffuScene and LayoutGPT. We will add this table and discussion to the revised paper.
>
> | On 1 TitanRTX | Generation time per layout  | API Calls |
> |---------------|-----------------------------|-----------|
> | DiffuScene    | 27.43s                    | No        |
> | Sync2Gen      | 0.10s                     | No        |
> | ATISS         | 0.41s                     | No        |
> | LayoutGPT     | 10.42s                    | Yes; o1   |
> | Ours          | 25.52s                    | Yes; o1   |
>
> **Q: Thorough evaluation of intermediate results and each stage of generation.**
>
> We highlight that we provide a thorough quantitative analysis of each intermediate stage in our framework:
> 1. Library learning: we measure the diversity of functions and accuracy of compression in Table 2, showing large improvements over an inference-only baseline.
> 2. Program generation: we ablate our pipeline without the pose model to show the contribution of layout generation alone in Table 1.
> 3. Program execution: as this step uses deterministic Python code, we assume exact execution.
> 4. Pose prediction: we evaluate orientation predictions of primary and dependent objects in Table 3, showing that conditioning on program structure improves dependent objects’ predictions.
> 5. Full scene generation: we evaluate via quantitative metrics in Table 1, as well as a human study in the main text.
>
> Thank you for your suggestion, we will revise the main text to clearly highlight these intermediate evaluations and discuss their performance.
>
> **Q: Generalization to scenarios other than lived-in scenes.**
>
> Yes, FactoredScenes can generalize to non-lived-in scenes (e.g., designed or staged rooms) with just our program generation step alone, as shown in the wo/ poses variant in Table 1. For generalization to entirely different domains (e.g., outdoor scenes), pose prediction would require retraining on new orientation data, and potentially our learned library would need to be augmented with new wake-sleep stages (without requiring handcrafted functions).
>
> However, we expect the distributions learned from ScanNet to transfer well across many indoor scene types. To showcase performance within indoor domains, we include new results on the office category. We follow the same evaluation protocol as in the main paper and compute FID and KID between generated layouts and ScanNet office scenes.
>
> |      | Office FID | Office KID |
> |------|------------|------------|
> | Ours | 88.50    | 0.032     |
>
> **Q: Ability to capture inherent structures compared to existing works.**
>
> Compared to existing LLM-based works which rely on handcrafted and fixed libraries, FactoredScenes learns its library from example rooms, enabling discovery of reusable, data-driven structural abstractions underlying scenes. FactoredScenes’ LLM generations are constrained via this learned library, and we hypothesize that this learnt structural prior helps reduce unnatural generations by anchoring the generation space to abstractions learned from example rooms.
>
> Compared to non-LLM baselines, FactoredScenes explicitly models discrete, hierarchical structure, which we hypothesize improves generalization to novel room layouts, as we see strong performance even in the no-pose setting. Thank you for your feedback, we will add these discussions in the revised text.

---

> > ### Comment · Reviewer_ziVh · 2025-08-03
> >
> > I appreciate the authors' clarifications and additional comparisons. My main concerns have been addressed by the rebuttal. I would lean to accept the paper by involving the additional evaluations and discussions in the revised version.

---

> > > ### Author Response · Authors · 2025-08-04
> > >
> > > Thank you again for the helpful and constructive feedback. We will make sure to incorporate all additional evaluations and discussions into the revised paper.

---

### Official Review · Reviewer_ME6t · 2025-07-03

**Clarity:** 3
**Significance:** 2
**Originality:** 3
**Rating:** 5
**Confidence:** 4

**Summary:**

This paper proposes the FactoredScenes framework, which synthesizes realistic 3D real-world scenes by learning layout structures from synthetic data via library learning and predicting object poses with a program-conditioned model. The approach decomposes scene generation into hierarchical components—program generation, layout execution, pose prediction, and object retrieval—demonstrating significant improvements over state-of-the-art methods in generating ScanNet-like scenes, though it retains limitations in program logic and data processing.

**Questions:**

1. Does the LLM-driven library learning process explicitly differentiate between scene types (e.g., bedrooms vs. living rooms) when capturing structural patterns? Are there experiments validating this differentiation?
2. Given the model is trained on only 707 ScanNet scenes, how is overfitting mitigated?
3. The paper mentions "commonsense rules" for program generation, but how are these constraints formalized or quantified? Are there ablation experiments isolating their impact on scene realism?
4. For non-axis-aligned objects, does the pose model consider functional geometric dependencies (e.g., monitor orientation relative to seating)? Current descriptions focus on hierarchical dependencies but lack clarity on geometric reasoning.
5. Do the experimental comparisons include 2024–2025 advances in 3D scene generation (e.g., DiffInDScene, Blockfusion)? The baselines listed may not reflect the latest state of the art.

**Ethical Concerns:**

["NO or VERY MINOR ethics concerns only"]

**Final Justification:**

The author seems to have addressed my concerns. However, since this submission was randomly assigned to me and it is not my area of expertise, despite my best efforts in reviewing, I still suggest that the AC take into account the opinions of other reviewers more. Thank you.

**Limitations:**

yes

**Quality:**

3

**Strengths And Weaknesses:**

**Strengths**
1. The factored representation decomposes scenes into structural programs and pose variations, leveraging synthetic 3D-Front data for library learning and LLMs for program generation.
2. Experiments demonstrate substantial improvements.
3. The paper is well-written, clearly articulating the problems to be solved and the problem-solving approach.

**Weakness**
1. The paper appears to overly rely on LLM programs for generation, which may lead to catastrophic generation results when unnatural object positions (such as nightstands in the middle of a bed) occur.
2. The proposed method does not address the overlap issue in scenes.
3. LLM inference requires high computational resources, and I am unsure whether the generative quality will be affected in other scenarios (e.g., offices).
4. Although the paper claims to compare multiple methods, it lacks quantitative comparison experiments (such as KL, FID, SCA, etc.).

---

> ### Author Rebuttal · Authors · 2025-07-30
>
> Thank you for your constructive feedback, which has greatly improved our paper.
>
> **Q: Quantitative comparisons.**
>
> We clarify that our main quantitative evaluation includes FID and KID, reported in Table 1 of the main paper, across both bedroom and living room scenes. Thanks to your suggestion, we have now included KL divergence over object category distributions, following prior work (e.g., ATISS, DiffuScene). We see that FactoredScenes comfortably outperforms prior works on this metric across both scene types (lower is better), indicating stronger alignment with real-world category distributions.
>
> |            | Bedroom KL | Living room KL  | Bedroom FID | Living room FID | Bedroom KID | Living room KID |
> |------------|------------|------------|-------------|------------|-------------|------------|
> | DiffuScene | 7.861     | 6.257     | 135.57      | 186.54     | 0.130       | 0.177      |
> | Sync2Gen   | 8.052     | 6.402     | 126.67      | 139.30     | 0.127       | 0.117      |
> | ATISS      | 7.916     | 6.295     | 120.27      | 176.95     | 0.117       | 0.166      |
> | LayoutGPT  | 7.663     | 6.284     | 109.40      | 157.69     | 0.102       | 0.142      |
> | Ours       | **7.476** | **6.127** | **67.51**   | **83.49**  | **0.020**   | **0.024**  |
>
> We also note that we provide a thorough quantitative analysis of each intermediate stage in our framework:
> 1. Library learning: we measure the diversity of functions and accuracy of compression in Table 2, showing large improvements over an inference-only baseline.
> 2. Program generation: we ablate our pipeline without the pose model to show the contribution of layout generation alone in Table 1.
> 3. Program execution: as this step uses deterministic Python code, we assume exact execution.
> 4. Pose prediction: we evaluate orientation predictions of primary and dependent objects in Table 3, showing that conditioning on program structure improves dependent objects’ predictions.
> 5. Full scene generation: we evaluate via quantitative metrics in Table 1, as well as a human study in the main text. We will revise the main text to clearly highlight these evaluations and discuss their performance.
>
> **Q: LLM-based generation.**
>
> Indeed, FactoredScenes programs rely on LLMs’ commonsense knowledge as part of the generation process, as many state-of-the-art prior works do (e.g., LayoutGPT, Holodeck, LayoutVLM). However, unlike these prior works, we learn the underlying library representing structure in scenes, and constrain LLM generations via this learned library. We hypothesize that this learnt structural prior helps reduce unnatural generations by anchoring the generation space to abstractions learned from example rooms, which leads to the strong performance of the FactoredScenes wo/ poses variant in Table 1.
>
> We agree that LLM-generated programs can occasionally yield unnatural scenes, which we highlight in the limitations section. Thanks to your suggestion, we have added experiments that test the validity of FactoredScenes’ LLM-generated programs, as well as the robustness of downstream results to potential errors in these programs.
>
> First, we report the percentage of FactoredScenes’s generated programs that execute successfully without errors and the percentage of programs that are semantically valid. As there is no ground truth for semantically valid programs, we define semantic validity using a conservative heuristic: a generated layout is deemed invalid if any pair of objects (excluding lamps and pillows) has IoU > 0.3. Here, we see that a high percentage of programs execute successfully, and even with this conservative, semantically valid criteria, we see a high percentage of programs marked as valid, which indicates that catastrophic errors are rare.
>
> | Across 100 rooms | % of program that executes successfully | % of programs that are semantically valid |
> |------------------|-----------------------------------------|-------------------------------------------------|
> | Ours             | 97%                                     | 90%                                             |
>
> Second, we evaluate the effect of FactoredScenes’ program validity on downstream generation results, and report performance for scenes generated from valid vs. invalid programs. As shown below, semantically valid programs as per our heuristic produce lower (better) performance, although the difference is slight.
>
> | Generation | KID of semantically **valid** programs | KID of semantically **invalid** programs |
> |-----------------------|--------------------------------|------------------------------|
> | Ours                  | 0.0183                         | 0.0192                       |
>
> Lastly, we investigate the robustness and sensitivity of our program generation to different prompt variations, under different few-shot prompting configurations (5-shot, 3-shot, 1-shot). FactoredScenes shows consistently strong performance across variants, significantly above that of prior works, demonstrating its robustness in generation.
>
> | Robustness        | All           | 5-shot FID | 3-shot FID | 1-shot FID |
> |-------------------|---------------|------------|------------|------------|
> | Ours bedroom      | 64.86 ± 4.36 | 67.51      | 59.83      | 67.25      |
> | Ours living room  | 91.26 ± 6.76 | 83.49      | 95.74      | 94.55      |
>
> We thank you for the feedback and will incorporate these new analyses into the revised paper.
>
> **Q: Generation quality in other scenarios (e.g., offices).**
>
> As suggested, here, we add additional results of FactoredScenes generating offices, a new room category. We follow the same evaluation protocol as in the main paper and compute FID and KID between generated layouts and ScanNet office scenes.
>
> |      | Office FID | Office KID |
> |------|------------|------------|
> | Ours | 88.50    | 0.032     |
>
> **Q: Modeling of geometric dependencies.**
>
> Our core hypothesis is that functional geometric relations stem from intentional semantic relations, and they can be captured via our framework. In cases like monitor orientation relative to seating, FactoredScene can encode such dependencies, as it arises from semantics (e.g., seat must face monitor, hence the generation should be dependent), which are then used by our pose model to predict geometry. However, we do not explicitly model purely geometric reasoning. We agree that incorporating these geometric constraints would be an interesting future direction.
>
> **Q: Differentiation of scene types in the library learning process.**
>
> FactoredScenes' LLM-driven library learning process does not explicitly differentiate between scene types, and captures underlying structure across all room designs in 3D-Front. This encourages the discovery of general, reusable structural abstractions that are applicable across diverse, open-world rooms.
>
> However, during generation, FactoredScenes does condition on the target scene type, which is passed as part of the prompt to the LLM, and hence the LLM chooses appropriate functions to use. We analyze the distribution of library function usage across room categories in the generated program, and see that, for example, the function parallel is used more in living rooms, while cluster_placement is used more in bedrooms. Such insights are valuable as they help us characterize structural differences between scene types.
>
> **Q: Comparison to DiffInDScene and Blockfusion.**
>
> DiffInDScene and Blockfusion focus on unconditional geometric generation, producing SDF volumes that are post-processed into 3D meshes (via marching cubes), without recovering semantic object-level layouts. To render scene visualizations, they use off-the-shelf tools to directly add texture to their scene geometry (e.g., DreamSpace, Meshy). We note that Blockfusion additionally allows users to specify a 2D layout as input, not output.
>
> In contrast, FactoredScenes targets unconditional room generation via semantic scene layouts. We compare FactoredScenes against top representative works that solve this task, spanning different model classes: ATISS (autoregressive), DiffuScene (diffusion), Sync2Gen (VAE), and LayoutGPT (LLM-based). We will clarify this distinction in the main text and add additional discussion in the revised paper.
>
> **Q: Addressing the overlap issue.**
>
> Currently, FactoredScenes does not explicitly address the overlap issue in scenes. Thanks to your suggestion, we follow prior work (e.g., LayoutVLM), and add a manual overlap checker in the final stage of our method, applied after pose predictions. Scenes with any pair of objects (excluding lamps and pillows) of IoU > 0.3 are filtered out. We see that this rejection sampling step improves layout quality across bedrooms and living rooms, and we will include this variant and its results in the revised paper.
>
> |                  | Bedroom KID | Living room KID |
> |------------------|-------------|------------|
> | Ours             | 0.020       | 0.024      |
> | Ours wo/ overlap | 0.016       | 0.023      |
>
> **Q: Overfitting mitigation.**
>
> We highlight that we design FactoredScenes as a structured framework, precisely as fully end-to-end methods tend to overfit on small datasets like ScanNet. For the only component of FactoredScenes trained on ScanNet, the pose prediction model, we intentionally keep the architecture lightweight, limiting model capacity to reduce overfitting, as indeed, the primary constraint here is the availability of real-world datasets.
>
> **Q: Clarification of commonsense rules.**
>
> We clarify that FactoredScenes does not define explicit commonsense rules. Instead, we use LLMs as a repository of implicit commonsense knowledge, from which we can generate programs when guided by our learned library. Given that LLMs are trained on extensive corpora of human data, we hypothesize that they contain commonsense knowledge about object placement in rooms for a wide range of room types. Thank you for the feedback. We will clarify this in the main text.

---

> > ### Author Response · Authors · 2025-08-07
> >
> > As the discussion period continues, please don’t hesitate to reach out if any additional information or clarification would be helpful. Thank you again for your constructive feedback.

---

### Note · Authors · 2025-08-12

We thank the reviewers for their constructive comments and engagement during this process. From the discussions, reviewers expressed satisfaction with our updates, and their feedback all appeared positive toward our revised work. In response to their suggestions, we have addressed all raised points with experiments, analyses, and clarifications.

We added additional evaluations of KL divergence to complement FID/KID, reported generation results on a new *office* category to demonstrate generalization within indoor scenes, conducted 3D-Front experiments without the real-world component for baseline comparison, and highlighted quantitative analysis of each intermediate stage in FactoredScenes.

We reported execution success and semantic validity rates for generated programs, analyzed downstream performance for valid vs. invalid programs, and tested prompt robustness across 1/3/5-shot settings with consistently strong results. We also added runtime comparisons, implemented an overlap-removal variant following prior work, and clarified details surrounding the full FactoredScenes pipeline.

Following these updates, reviewers acknowledged that the rebuttal addressed their concerns, with several noting they would raise their rating or lean toward acceptance. Building on this feedback, we will comprehensively incorporate new evaluations and discussions into our revised text.

In summary, FactoredScenes introduces a library-learning approach that constrains LLM-based program generation with underlying room structure, and proposes a program-conditioned model that leverages hierarchical dependencies for realistic pose predictions. Our modular framework achieves significant improvements over prior works in generating real-world scene layouts, and we believe our revised work will make a meaningful contribution to the NeurIPS community.

We appreciate the AC’s consideration, and thank the reviewers for their time and effort.

---

### Decision · Program_Chairs · 2025-09-17

**Decision:**

Accept (poster)

**Comment:**

The paper presents a method for generating 3D scenes by generating layouts and object poses using program synthesis with LLMs . The reviewers appreciated the method and the results, but initially raised concerns over limited evaluations and differences with the state of the art. The rebuttal addressed these concerns very well, and all reviewers are in favor of acceptance. The paper advances this problem by being able to synthesize real-world scenes with noisy data.